# Recent Advances in Ultrasound-Guided Peripheral Intravenous Catheter Insertion

**DOI:** 10.3390/nursrep15100359

**Published:** 2025-10-08

**Authors:** Amélie Bruant, Laure Normand

**Affiliations:** 1Department of Intensive Care Medicine, Centre Hospitalier de Melun-Senart, 77000 Melun, France; amelie.bruant@ghsif.fr; 2Department of Intensive Care Medicine, Centre Hospitalier de Fontainebleau, 77300 Fontainebleau, France

**Keywords:** peripheral intravenous catheter, ultrasound, difficult venous access

## Abstract

**Background/Objectives:** This narrative review addresses ongoing controversies and advancements concerning ultrasound-guided peripheral intravenous (IV) catheter insertion, and the impact of ultrasound guidance on success rate, procedural time, patient and staff experience, complications and costs, as well as requirements for its use. **Methods:** A literature review was conducted.
**Results:** Growing evidence suggests that ultrasound-guided insertion of peripheral IV catheter represents a superior technique across various patient populations, particularly those presenting with difficult IV access (DIVA). Key findings highlight significant improvements in first-attempt success rates, reduction of procedural complications, and enhanced patient comfort. Ultrasound-guided insertion is also associated with an increase in catheter dwell time, a reduction in repeat procedures and in central line placements, leading to improved resource utilization and the potential for substantial long-term cost-effectiveness, despite the cost of initial investment and training. However, obtaining these improvements involves a critical importance for standardized training, adherence to rigorous aseptic techniques, and generalization of the transformative impact of ongoing technological advancements in ultrasound devices. **Conclusions:** The collective body of evidence supports the widespread adoption of ultrasound-guided peripheral IV cannulation as an evidence-based best practice in modern healthcare.

## 1. Introduction

Peripheral IV catheter insertion remains one of the most frequently performed invasive procedures in healthcare settings globally [1,2,3]. Despite its commonality, traditional “blind” methods, which rely solely on palpation and visualization of veins, are frequently associated with unacceptably high failure rates, up to 40% according to available studies [2]. These challenges often necessitate multiple insertion attempts, leading to increased patient anxiety, distress, and potential avoidance of essential medical treatments [4]. Beyond patient discomfort, repeated unsuccessful attempts consume considerable nursing time, increase pressure on healthcare staff, and incur higher medical costs due to prolonged procedures and potential delays in initiating critical therapies [5,6,7].

Patients with specific anatomical variations, deeply situated, fragile blood vessels, or a history of difficult IV access (DIVA) are particularly susceptible to these challenges [8]. Vulnerable demographics, such as pediatric patients with inherently small veins, obese individuals with deep tissue layers, and elderly patients with fragile or “rolling” veins, commonly experience heightened difficulty in obtaining reliable vascular access [9]. The cumulative burden of these frequent failures, encompassing patient suffering, staff inefficiency, and economic strain, represents a significant, often underestimated, systemic cost [10]. This comprehensive impact extends beyond direct, per-procedure expenses to affect overall hospital efficiency and patient flow.

Ultrasound guidance offers a transformative solution to these long-standing issues by enabling real-time visualization of the target vein and surrounding anatomical structures. The clear visualization provided by ultrasound directly addresses the limitations of traditional methods, offering a proactive, evidence-based approach to vascular access.

While the provided literature does not detail a specific historical timeline, it indicates a rapid and significant integration of ultrasound into vascular access procedures [11,12]. Initially regarded as an adjunct tool to assist in challenging cases, ultrasound-guided peripheral IV catheter insertions have evolved into a widely recognized and, in many contexts, a standard of care approach, particularly for DIVA.

Given its increasing adoption in healthcare institutions, the aim of this comprehensive review of ultrasound-guided intravenous catheter insertion is to summarize its impact on success rate, procedural time, patient and staff experience, complications and costs, as well as requirements for its use. By critically appraising existing literature, this review would also highlight potential technological advancements and areas requiring further research.

## 2. Materials and Methods

This narrative review examined the current available evidence regarding ultrasound-guided peripheral intravenous cannulation. A literature review was conducted with a search strategy of observational and interventional studies, as well as meta-analyses and systematic reviews, about ultrasound-guided peripheral intravenous cannulation, in PubMed, Cochrane Library of Clinical Trials, and Google Scholar. Manuscripts in the English language published from January 1, 2010 until May 31, 2025 were included. The early light ultrasound machines were hampered by poor image quality, but imaging quality increased from 2010. For this reason, only studies published in the last fifteen years were included.

The primary search criteria for PubMed and Google Scholar included “peripheral intravenous cannulation”, “peripheral intravenous access”, “peripheral intravenous catheterization”, and were connected by the Boolean “AND” with the terms “ultrasound”, “ultrasound guidance”, “ultrasonography”, or ultrasound-guided” (see Appendix A). For Cochrane Library, the terms “Ultrasonography” and “Catheterization, Peripheral” were used.

This review focuses on the following outcome measures: success rate, procedural time, patient and staff experience, complications, and costs.

Studies describing the success rate, procedural time, patient and staff experience, complications, costs, training and technological advancements as well as recommendations from key organizations for ultrasound-guided peripheral IV cannulation, were included.

Studies were excluded for the following reasons: intravenous insertion of other devices such as central venous catheters, peripheral inserted central venous catheters, dialysis catheters, and arterial catheters.

The two authors screened eligible studies independently, and classified them as being relevant or not relevant. Secondly, the full text of the articles that were classified as being relevant was analyzed by both authors independently. Hereafter, both authors decided individually if the study was significant for the outcomes, without standardized assessment tools. Any discrepancy between the authors was resolved by a common agreement.

Records identified through database searching were 116 from PubMed, 9 from Cochrane Library and 21 additional records from Google Scholar. After screening for the outcome measures of interest and guidelines, 44 manuscripts were included in this qualitative review (See Figure 1).

## 3. Results

### 3.1. Clinical Efficacy and Patient Outcomes

#### 3.1.1. Enhanced Success

Recent studies consistently demonstrate the marked superiority of ultrasound guidance in achieving higher first-attempt and overall success rates for IV catheter insertion. The EPIC randomized trial provides compelling evidence from a pediatric population [13]. This trial reported a first-time insertion success rate of 85.7% for ultrasound-guided peripheral IV catheter insertion, a significant improvement compared to the 32.5% achieved with standard techniques (*p* < 0.001). Importantly, this enhanced success was observed across all categories of insertion difficulty (low, medium, and high risk), suggesting a broad applicability beyond only the most challenging cases.

For adult patients experiencing difficult peripheral IV access, a randomized crossover study further substantiated these benefits [14]. This study found that the first-attempt success rates were 63.9% in the ultrasound-guided group, versus 13.9% with standard techniques (OR 10.97, 95% C.I. 3.43 to 35.13, *p* < 0.0001). The overall success rates were respectively 88.9% and 13.9% (OR 49.60, 95% C.I. 12.18 to 202.05, *p* < 0.0001). Similarly, a training program for registered nurses, which implemented longer-length peripheral IV catheters under ultrasound guidance for DIVA patients, achieved an impressive 95% first-stick success rate [15], which was attributed to enhanced visualization.

Systematic reviews and meta-analyses consistently corroborate these findings. One meta-analysis about peripheral IV cannulation in emergency and trauma patients found a two-fold increase in the odds of first-pass success when using ultrasound guidance (OR 2.1; 95% C.I. 1.65 to 2.7; *p* < 0.001) [16]. Another comprehensive meta-analysis of 15 studies on DIVA patients concluded that Point-of-Care Ultrasound (POCUS) significantly improved overall success rates (OR 4.25, 95% C.I. 2.01 to 8.98, *p* = 0.0001) and first-attempt success (OR 5.31, 95% C.I. 2.57 to 10.96, *p* < 0.0001) [17]. This was coupled with fewer skin puncture attempts (mean difference −0.83; 95% C.I. −1.32 to −0.33; *p* = 0.001).

In a recent systematic review including children and adults in operating rooms, emergency departments, and intensive care units, ultrasound guidance was also associated with a higher first-attempt success rate (OR 3.07; 95% C.I. 1.66 to 5.65; *p* < 0.001) [18]. On average, traditional IV catheter placement typically requires between 1.5 and 2.5 insertion attempts, whereas ultrasound guidance reduces this to a range of 1.1 to 1.7 attempts. The consistent and substantial improvements in first-attempt success rates across diverse patient groups underscore the profound superiority of ultrasound-guided peripheral IV cannulation over traditional methods. The data suggest that if ultrasound-guided peripheral IV cannulation is demonstrably superior even for “easy” cases, it fundamentally questions the continued reliance on palpation and visualization as the default method for any peripheral IV catheter insertion. This implies, at least in high-income countries, a potential future-oriented shift in standard practice for all peripheral IV catheter insertions, leading to universally improved patient outcomes and procedural efficiency.

#### 3.1.2. Procedural Efficiency

Beyond enhancing success rates, ultrasound guidance consistently improves procedural efficiency by reducing the number of insertion attempts and shortening the overall time required for cannulation. The DIAPEDUS study, published in 2024 and focusing on pediatric DIVA patients, reported a significantly lower mean procedural time of 2.7 ± 2.2 min with ultrasound guidance, compared to 10 ± 6.4 min for the standard technique (*p* < 0.0001) [19]. For children needing an IV vascular access in the emergency department, a randomized study reported a significantly shorter time from randomization to intravenous line flush using ultrasound-guided catheter insertion (median 14 min versus 28 min) [20]. A meta-analysis on DIVA patients found that ultrasound-guided peripheral IV catheter insertion was associated with shorter procedure time (mean difference −9.75 min; 95% C.I. −15.44 to −4.06; *p* = 0.0008) [17]. Cannulation using bi-plane imaging seems to be even faster compared to mono-plane ultrasound methods [21].

#### 3.1.3. Patient and Staff Experience

The high first-attempt success rates, reduced number of attempts, and faster procedural times are not isolated benefits [22]; they form a synergistic relationship that profoundly impacts patient comfort and satisfaction [23], minimizes complications, and significantly improves overall workflow efficiency. When initial attempts are successful, the need for multiple attempts decreases, directly leading to less patient discomfort and anxiety typically associated with multiple failed cannulation attempts. This efficiency also reduces the workload on nursing staff.

Enhanced patient comfort and satisfaction are widely highlighted as key advantages of ultrasound-guided IV access [24,25,26]. By minimizing the number of painful needle sticks, ultrasound-guided peripheral IV cannulation contributes to a less traumatic healthcare experience, leading to vessel health preservation.

#### 3.1.4. Complication Profile

Ultrasound guidance plays a crucial role in mitigating complications associated with IV access due to its ability to provide real-time visualization and precise needle placement. A meta-analysis of 8 studies on adult patients indicated fewer complications, specifically for vascular injury (OR 0.15; 95% C.I. 0.07 to 0.32; *p* < 0.00001), and hematoma formation during the puncture process (OR = 0.24; 95% C.I. 0.08 to 0.69; *p* = 0.008) [27].

#### 3.1.5. Dwell Time

Regarding catheter dwell time, longer-length peripheral IV catheters placed under ultrasound guidance have demonstrated impressive longevity, with dwell times ranging from 1 to 80 days and an 83.6% treatment completion rate using only one catheter [15]. In children, ultrasound-guided intravenous cannulation is also associated with a significantly longer dwell time (median 7.3 days [95% CI 3.7 to 9.5] versus 2.3 days [95% CI 1.8 to 3.3]) [20]. Thus, precise ultrasound-guided placement can contribute to improved catheter survival, further reducing the need for repeat procedures and associated complications.

### 3.2. Application in Specific Patient Populations

Difficult Intra-Venous Access patients

Ultrasound guidance is strongly advocated as a primary approach for patients identified with DIVA. These patients are disproportionately affected by traditional methods, often enduring multiple failed attempts, increased discomfort, and critical delays in receiving medical treatments [6,7]. Ultrasound-guided insertion has been consistently shown to significantly improve first-attempt success rates in this challenging cohort [18].

A significant benefit of successful ultrasound-guided peripheral IV cannulation in DIVA patients is the reduction in the need for more invasive central venous catheters (CVCs), which are associated with higher risks, greater costs, and specialized personnel requirements. One study highlighted a reduction of 85% in the need for CVCs in non-critical patients through the implementation of a program for ultrasound-guided peripheral IV cannulation [28]. The use of longer-length peripheral IV catheters under ultrasound guidance has also demonstrated remarkable efficacy in DIVA patients, achieving a 95% first-stick success rate, which translates to a better patient experience and increased nursing efficiency [15].

Pediatric Patients

Peripheral IV catheter insertion in children presents unique challenges due to their smaller, more fragile veins and the heightened emotional impact of multiple attempts, resulting in anxiety and distress for both children and their families. Ultrasound-guided peripheral IV catheter insertion is recognized as a superior alternative for this vulnerable population. The EPIC study found that ultrasound-guided insertion dramatically improves first-time cannulation success in children (85.7% vs. 32.5% for standard technique) [13], supporting the widespread adoption of ultrasound-guided cannulation in this setting.

The DIAPEDUS study, specifically focusing on pediatric DIVA patients, further reinforced these benefits, showing a 90% first-attempt success with ultrasound-guided techniques versus 18% for standard methods, coupled with reduced procedural time and fewer attempts overall [19]. A retrospective study published in 2024 found that ultrasound-guided insertion in children resulted in longer dwelling times and fewer complications compared to blind insertion [29]. These consistent improvements highlight how ultrasound-guided cannulation addresses the specific anatomical and emotional challenges inherent in pediatric vascular access. Table 1 summarizes the main studies cited in this review.

Emergency and Trauma Settings

Point-of-Care Ultrasound (POCUS) has emerged as an indispensable tool for peripheral IV catheter placement in emergency and trauma departments, where rapid and reliable vascular access is critical. It consistently improves first-attempt success rates, decreases the number of cannulation attempts, and enhances overall procedural efficiency [16]. In time-critical scenarios, traditional methods can lead to dangerous delays and increased complications [6,7]. POCUS helps overcome these challenges by providing immediate visualization [17]. However, it is important to note that for critical patients requiring immediate, emergent access (e.g., cardiac arrest, severe trauma), traditional peripheral IV or intraosseous access may still be more appropriate initial choices due to severe time and organization constraints. Once immediate life-saving interventions are underway and the patient is relatively stabilized, ultrasound-guided peripheral IV cannulation becomes the preferred method for more stable, difficult access patients. The effectiveness of POCUS in these high-pressure environments is further enhanced by nurse-led POCUS training programs, which have demonstrated high post-training success rates, often exceeding 90% [16].

### 3.3. Economic Implications and Cost-Effectiveness

The initial investment in ultrasound equipment and comprehensive training programs represents a notable upfront cost and studies on cost effectiveness of ultrasound-guided peripheral IV cannulation are few. Cost-effectiveness analyses typically compare the expenses associated with ultrasound machines and supplies against the costs incurred from difficult IV insertions, the treatment of complications, and the extensive nursing time consumed by traditional blind insertions.

An observational cohort study in the emergency department of an academic center in the USA found that ultrasound-guided peripheral IV cannulation was associated with a cost reduction of $83 (95% C.I. −$104 to −$62; *p* < 0.001), compared to traditionally-placed catheters [30]. Ultrasound-guided insertion was also associated with an increase in catheter dwell time relative to hospital length of stay [30]. In DIVA patients, the average total cost for ultrasound-guided long peripheral catheter insertion was lower, compared to traditionally placed catheters [31]. In the United Kingdom, the cost of long peripheral IV catheters placed under ultrasound guidance was also reported to be £89.22 lower compared to a midline insertion, alongside a reduction of 1 kg in clinical waste per procedure [15].

The EPIC trial noted that ultrasound-guided peripheral IV cannulation had higher immediate healthcare costs (US$ 5.83 per person) [13]. However, this immediate cost increase must be weighed against the long-term savings and improved patient outcomes. The overall economic benefit stems from preventing downstream complications, reducing repeat procedures, and optimizing staff time, leading to long-term cost savings and improved resource utilization. The long-term value, encompassing both financial and patient-centered benefits, could outweigh the marginal immediate cost increase.

The reduction in central line placements due to successful ultrasound-guided peripheral IV cannulation for DIVA patients represents a significant strategic advantage for healthcare systems. Central lines are more invasive, require specialized staff such as radiologists or anesthesiologists, and are inherently more costly and prone to severe complications. If ultrasound-guided peripheral IV cannulation can effectively provide venous access for a large proportion of DIVA patients who would otherwise require a central line, it reduces the demand for these higher-acuity procedures. This means fewer patients undergo higher-risk interventions, and valuable specialist time and high-cost resources, such as operating rooms or radiology suites, can be reallocated to other critical needs. This systemic benefit, beyond individual patient outcomes, optimizes overall hospital resource management.

Investing in nurse-based ultrasound-guided programs is a critical component of achieving cost-effectiveness. This approach decentralizes a specialized skill, reduces reliance on higher-cost providers such as radiologists and anesthesiologists, and improves departmental workflow. A Canadian study found that ultrasound-guided insertion of peripheral catheters is a promising cost-saving approach [32]. This study demonstrates that the economic benefits are not solely derived from the technology itself, but significantly from who performs the procedure. Empowering a broader base of healthcare professionals through structured training is key to maximizing efficiency and reducing overall system costs by distributing the workload, improving access, and reducing reliance on more expensive, specialized personnel or procedures.

Successful integration of ultrasound-guided IV cannulation programs into healthcare settings requires careful consideration of several factors. These include establishing appropriate credentialing standards for healthcare professionals, including nurses, defining comprehensive training requirements, managing the costs of equipment and supplies, evaluating the impact on departmental workflow efficiency, and ensuring high levels of patient satisfaction. Thoughtful and strategic implementation is therefore crucial to ensure both the feasibility and the ultimate cost-effectiveness of these programs.

While the existing literature provides valuable insights into the cost-effectiveness of ultrasound-guided insertions, certain limitations should be acknowledged such as variability in study designs (methodologies, patient populations, healthcare settings); short follow-up periods (many studies do not account for long-term outcomes which may affect the overall assessment of cost-effectiveness) and limited geographic scope. The cost-effectiveness of ultrasound-guided insertions has not been evaluated in resource-limited countries and continental Europe public healthcare systems.

### 3.4. Clinical Guidelines and Best Practices

#### 3.4.1. Recommendations from Key Organizations

Medical and nursing organizations have increasingly integrated ultrasound guidance into their clinical guidelines for IV access, particularly for patients with DIVA. The Infusion Nurses Society (INS), in its 2024 Infusion Therapy Standards of Practice, explicitly supports the use of visualization technologies [33]. It specifically recommends ultrasound guidance insertions in adults with DIVA, citing Level I evidence for its ability to reduce insertion attempts and decrease reliance on more invasive central vascular access devices.

The guidelines of the American College of Emergency Physicians provide detailed recommendations for optimal scanning techniques, general procedure setup, and injection precautions for ultrasound-guided vascular access [34].

The American Society of Echocardiography (ASE) published an updated guideline in February 2025, offering comprehensive, evidence-based recommendations on ultrasound-guided vascular cannulation procedures [12]. This guideline emphasizes three fundamental roles of ultrasound during vascular access: precannulation vessel assessment, dynamic ultrasound guidance, and identification of local complications. The Society of Hospital Medicine also recommends that providers use real-time ultrasound guidance for peripheral IV line placement in DIVA patients [35]. These guidelines highlight a shift in focus from whether to use ultrasound-guided peripheral IV cannulation to how to use it safely, efficiently, and consistently, emphasizing protocol development and specific techniques.

#### 3.4.2. Training and Competency Standards

The effectiveness and safety of ultrasound-guided peripheral IV cannulation are heavily reliant on rigorous and standardized training programs for healthcare professionals. Training programs typically incorporate didactic (classroom) instruction, hands-on simulation, and supervised clinical practice [15]. These programs aim to provide participants with the knowledge necessary to recognize, assess, and manage DIVA conditions, and to proficiently use ultrasound for vein visualization to ensure safe and accurate line placement [36].

Evidence supports the effectiveness of structured training: nurse-led vascular ultrasound training programs have consistently achieved high post-training success rates, often exceeding 90% [16,37]. In the DIAPEDUS study, nurses and pediatricians had completed departmental training, including at least 20 ultrasound-guided IV placements, prior to the study [19]. Comprehensive training bundles are available, covering essential aspects such as vein anatomy, appropriate catheter selection, adherence to aseptic technique, detailed insertion procedures, and validation of successful placement. These programs are designed not only to train individual practitioners but also to establish and maintain highly trained teams and to develop future trainers and preceptors within healthcare organizations. Experience plays a significant role in procedural efficiency; studies indicate that experienced operators achieve shorter procedure times with ultrasound, whereas inexperienced providers may initially require more time, underscoring the importance of adequate practice and competency development [38]. The effectiveness and cost-effectiveness of ultrasound-guided peripheral IV cannulation are not solely dependent on the technology itself but are profoundly influenced by the quality and standardization of training, ongoing competency, and rigorous adherence to best practices.

#### 3.4.3. Aseptic Technique

Maintaining strict aseptic technique is paramount to minimizing the risk of contamination and subsequent infection for peripheral IV catheter procedures. The INS Standards’ adoption of the Aseptic Non-Touch Technique (ANTT) framework for all vascular access procedures, including ultrasound-guided insertions, highlights its critical importance [33]. A specific patient safety risk associated with ultrasound-guided peripheral IV catheter is the potential for contamination from the ultrasound probe and gel if appropriate safety guidelines are not rigorously followed. The adoption and adherence to the Aseptic Non-Touch Technique is fundamental to enable prolonged dwell time for peripheral IV catheters inserted using ultrasounds.

### 3.5. Technological Advancements in Ultrasound for IV Access

Vascular ultrasound technology has undergone significant advancements, moving beyond early simple black-and-white images to sophisticated capabilities that greatly enhance visualization for IV access. Modern systems now offer color flow imaging, which provides crucial information on the speed and direction of blood flow, in addition to detailed anatomical shapes of blood vessels [39]. Some newer systems incorporate 3D imaging, providing more comprehensive information, especially beneficial for challenging cases. These technological improvements lead to delivering outstanding resolution, a larger field of view, decreasing artifacts, and defining vasculature with a 3D-like appearance, contributing collectively to superior image quality and enhanced insights during cannulation.

A notable trend in ultrasound technology for vascular access is the increasing prevalence of portable and wireless solutions, which significantly impact accessibility and workflow. Handheld ultrasound machines and wireless probes are gaining popularity due to their compact size and easy functionality, enabling rapid image acquisition directly at the point of care [40]. Wireless transducers connect seamlessly to mobile devices (smartphones, tablets) via Bluetooth or Wi-Fi, offering unparalleled accessibility, portability, real-time imaging capabilities, and improved ergonomics for the clinician. A key advantage of wireless systems is their enhanced ease of wrapping in sterile covers, which can improve adherence to maximal barrier precautions and reduce the risks of cross-contamination during sterile procedures [41]. These advancements in portability, wireless connectivity, and user-friendly interfaces are democratizing ultrasound use. This makes the technology more accessible to a wider range of healthcare professionals beyond traditional specialists, enabling a broader base of healthcare givers to perform ultrasound-guided peripheral IV cannulation and contributing to the economic and efficiency benefits observed in nurse-led programs.

The field of ultrasound-guided IV access is poised for further innovation with the development of advanced and automated technologies. Research is progressing on image-guided robotic systems designed to enhance the accuracy of DIVA procedures [42]. These systems integrate multi-degree-of-freedom robotic arms with end-effectors for precise navigation and needle insertion, aiming to reduce reliance on operator skill and improve consistency. However, these systems are still under development and have been tested on a porcine model.

Innovative training platforms, using augmented reality, are leveraging to overlay simulated needles onto live ultrasound feeds [43]. This allows trainees to practice non-invasive needling with virtual image overlays, effectively developing hand-eye coordination, positioning accuracy, and overall proficiency in ultrasound-guided procedures with minimal risk. Artificial intelligence (AI) could assist not only in real-time guidance during procedures but also in enhancing training modules, providing automated feedback, or even predicting optimal access points [44]. These innovative training tools utilizing augmented reality and the future integration of AI are crucial for accelerating skill acquisition and standardizing competency, thereby addressing a key barrier to widespread adoption.

## 4. Identification of Areas for Future Research and Development

Recent literature shows a consistent advantage for ultrasound guidance in DIVA patients, with no large studies reporting inferior outcomes. However, ultrasound-guided insertion in non-DIVA patients has not been rigorously studied and the ultrasound-guided use in extreme emergency situations has not been evaluated. While the clinical and economic benefits of ultrasound-guided peripheral IV cannulation are increasingly clear, further rigorous research is needed to establish definitive cost savings across diverse outpatient populations and specific healthcare settings. This will provide more granular data for resource allocation decisions. Continued development and validation of advanced technologies, such as AI-assisted guidance, automated vein detection, and robotic systems, hold immense promise for further improving accuracy, reducing operator dependence, and expanding the reach of ultrasound-guided peripheral IV catheter insertions, particularly for extremely difficult cases.

Ongoing research into optimal training methodologies, including the role of simulation and augmented reality, and the establishment of robust, standardized competency assessment frameworks remain vital to ensure high-quality, consistent practice as ultrasound-guided peripheral IV cannulation becomes more ubiquitous across all levels of healthcare providers. Further long-term studies on catheter dwell times and complication rates across diverse patient cohorts and various catheter types, such as longer-length peripheral catheters, would further strengthen the evidence base regarding the sustained benefits of ultrasound-guided peripheral IV catheter insertion.

## 5. Conclusions

The recent body of literature overwhelmingly affirms the superior efficacy of ultrasound-guided peripheral IV catheter insertion compared to traditional palpation and visualization methods. This is particularly evident for challenging patient populations, including those with difficult venous access and pediatric patients. Ultrasound-guided peripheral IV cannulation consistently demonstrates significantly higher first-attempt success rates, reduced procedural time, fewer insertion attempts, and a lower incidence of complications such as arterial puncture, nerve injury, and infiltration. These clinical improvements translate directly into enhanced patient comfort, reduced anxiety, and greater satisfaction, while simultaneously yielding long-term economic benefits for healthcare systems through improved efficiency, optimized resource utilization, and reduced downstream costs associated with failed attempts and complications. The widespread endorsement by leading medical societies further solidifies its position as an evidence-based best practice. The main barriers to implementing ultrasound-guided peripheral catheter insertion are initial equipment costs, training requirements, workflow integration and preservation of strict aseptic technique. The full realization of ultrasound-guided peripheral IV cannulation benefits requires a coordinated effort across technological innovation, comprehensive training programs, and supportive institutional policies and guidelines that facilitate its widespread, safe, and cost-effective adoption, as well as rethinking clinical workflows and highlighting the impact of ultrasound use on overall cost reduction for decision-makers.

## Figures and Tables

**Figure 1 nursrep-15-00359-f001:**
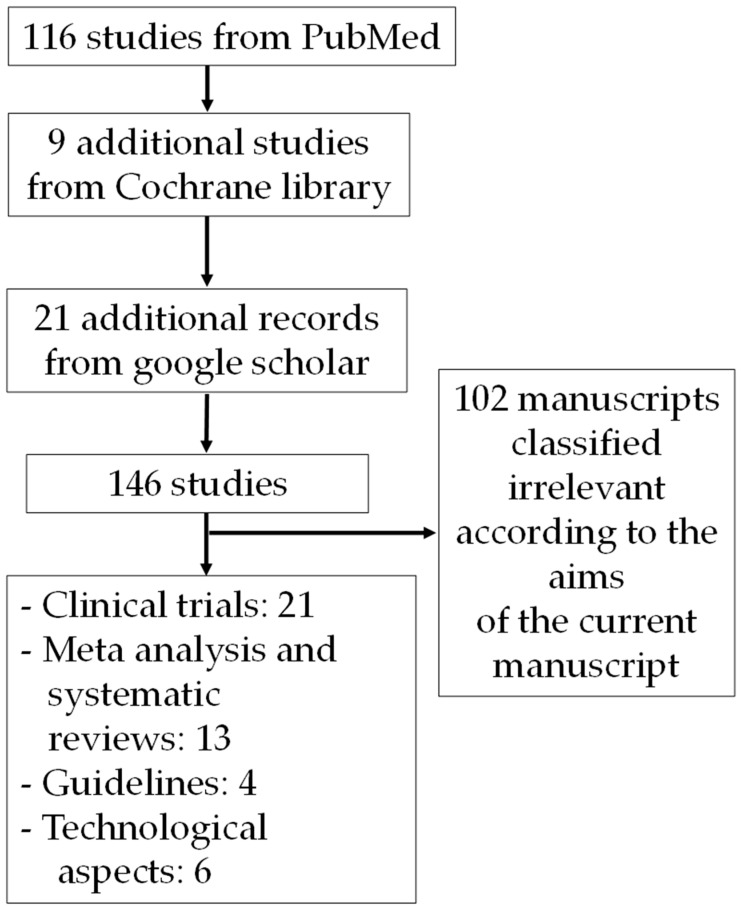
Flow diagram for the selection of studies.

**Table 1 nursrep-15-00359-t001:** Main studies on ultrasound-guided peripheral intravenous catheter insertion.

Study (First Author, Year)	Study Design	Population/Setting	Main Findings
Kleidon 2024 [13]	Randomized study	Children/pediatric hospital	First-attempt success rate of 85.7% for ultrasound-guided insertion vs. 32.5% with standard technique (*p* < 0.001). Higher immediate cost for ultrasound-guided insertion (+US$ 5.83 per person).
Kuo 2025 [14]	Randomized cross over study	Adults DIVA	First-attempt success rate of 63.9% for ultrasound-guided insertion vs. 13.9% with standard technique (*p* < 0.001).
Godfrey 2024 [15]	Observational, retrospective	Adults DIVA	Trained nurses using ultrasound insertion with longer-length peripheral IV catheters achieved 95% first-attempt success rate.
Elshikh 2025 [16]	Meta-analysis	Emergency and trauma patients	OR of first-attempt success with ultrasound guidance vs. standard technique: 2.1 (95% C.I. 1.65 to 2.7).
Zaki 2025 [17]	Meta-analysis	DIVA patients	OR of first-attempt success with ultrasound guidance vs. standard technique: 5.31 (95% C.I. 2.57 to 10.96). Shorter mean time of insertion using ultrasound guidance: −9.75 min [95% C.I. −15.44 to −4.06).
Poulsen 2023 [18]	Systematic review	Children and adults in operating rooms, emergency departments, and intensive care units	OR of first-attempt success with ultrasound guidance vs. standard technique: 3.07 (95% C.I. 1.66 to 5.65).
D’Alessandro 2024 [19]	Randomized study	Children DIVA, emergency department	Mean procedural time of 2.7 ± 2.2 min with ultrasound guidance, vs. 10 ± 6.4 min for standard technique (*p* < 0.0001).
Vinograd [20]	Randomized study	Children DIVA, emergency department	First-attempt success rate of 85.4% for ultrasound-guided insertion vs. 45.8% with standard technique, OR 1.9 (95% C.I. 1.5 to 2.4). Median procedural time of 14 min (95% C.I. 13 to 15) with ultrasound guidance vs. 28 min for standard technique (95% C.I. 21 to 31).
Hansel 2024 [22]	Randomized study	Adult hospitalized patients	First-attempt success rate of 90.2% for ultrasound-guided insertion vs. 35.7% with standard technique (*p* < 0.001). Median procedural time of 5 min with ultrasound guidance, vs. 10 min for standard technique (*p* < 0.001).
Salleras-Duran [23]	Randomized study	Adult emergency department DIVA patients	Higher satisfaction of patients with ultrasound-guided insertion.
Sengul [26]	Randomized study	Outpatient chemotherapy unit	Mean (standard deviation) pain score of 1.53 ± 1.13 for ultrasound-guided insertion vs. 2.96 ± 1.77 with standard technique (*p* < 0.05).
Xiong [27]	Meta-analysis	Adults	Comparison of ultrasound-guided insertion vs. standard technique, OR (95% C.I.) for failure rate 0.08 (0.04 to 0.16), for vascular injury 0.15 (0.07 to 0.32), for hematoma during the puncture process 0.24 (0.08 to 0.69).
Au [28]	Observational study	Adult DIVA patients in 2 emergency departments	Ultrasound-guided insertion prevented the need for central venous catheter placement in 85% of patients with difficult IV access.
Refosco [29]	Retrospective case/control study	Children DIVA, emergency department	Average duration of catheters inserted using ultrasound-guidance 5.3 ± 4.0 days with 66% of catheters removed electively vs. 2.5 ± 1.8 days and 30% for catheters inserted using standard technique (*p* < 0.001).

C.I.: confidence interval. OR: odds ratio.

## Data Availability

No new data were created or analyzed in this study. Data sharing is not applicable.

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
