# Peer review of "Recent Advances in Ultrasound-Guided Peripheral Intravenous Catheter Insertion"

_nursrep, 2025, doi:10.3390/nursrep15100359_

Round 1
Reviewer 1 Report
Comments and Suggestions for Authors
Thank you for the opportunity to evaluate this manuscript. While the topic is timely and the evidence base is promising, substantive issues in clarity, structure, and methodological transparency remain. I therefore recommend Major Revision, with the detailed points below to improve focus, readability, and scholarly rigor.
Abstract
The abstract provides a comprehensive overview of the review’s content. It succinctly summarizes key points such as the superiority of ultrasound-guided IV insertion in patients with difficult venous access, increases in first-attempt success rates, reductions in complications, improved patient comfort, and potential cost-effectiveness. Despite these strengths, the abstract could be improved for clarity and concision. The aims of the review are not explicitly stated. Adding an early sentence that clarifies the purpose would be helpful.
Introduction
The introduction effectively demonstrates the clinical importance of the topic. Although the content is relevant, the introduction could be presented with a clearer structure. To enhance readability, the following elements can be separated: first define the problem (as already done), then state explicitly the rationale for ultrasound guidance as the solution, and finally end with the review’s objectives. While the introduction covers the necessary background and significance, it should conclude with a concise statement of the review’s aims and scope.
Methods
The Methods section is generally well described and methodologically sound. For improvement, the authors are advised to clarify the nature of the review, add a flow diagram for the study-selection process, and state how study quality was (or was not) addressed. These additions would enhance the credibility and reproducibility of the review.
Identification of Areas for Future Research and Development
From a language standpoint, there is a grammatical error: “further rigorous researches are needed.” In this context, “research” is uncountable; it should be “further rigorous research is needed” or “studies are needed.” This should be corrected to conform to academic English.
Given the framing of “controversies” at the outset of this section, the discussion could explicitly acknowledge any counter-evidence or opposing viewpoints. For example, are there studies in which ultrasound does not show benefit, or situations where it may not be appropriate? The manuscript largely reports positive findings, which likely reflects the true balance of evidence. If there are no substantive recent contrary findings, this can be stated explicitly: “Notably, recent literature shows a consistent advantage for ultrasound guidance, with no large studies reporting inferior outcomes. Remaining debates center on cost, training, and implementation logistics rather than efficacy.” This would address expectations about “controversies” by clarifying that there is broad consensus on efficacy.
Conclusion
Section 5’s title includes “Future Directions,” but Section 4 already addresses future research; therefore, there is some overlap. It would be appropriate to revise the Conclusion to focus mainly on summary and practice recommendations, keeping future-research points brief or referring the reader back to Section 4 for details. As it stands, the Conclusion does not delve deeply into future directions (that is handled in Section 4); thus, removing “and Future Directions” from the heading could reduce confusion.
Author Response
We sincerely thank the reviewers for their useful comments to improve the manuscript.
Abstract. As suggested, the first sentence of the abstract was modified for more precisely state the aims of the manuscript.
Introduction. We modified the introduction chapter as requested, by first defining the problem, then indicating the rationale for ultrasound guidance and finally summarizing the aims of the manuscript.
Methods. A flow diagram was added to the manuscript. The quality of the studies was not addressed by a standardized assessment tool and this information was added to the chapter.
Identification of Areas for Future Research and Development. The sentence about ‘’further rigorous reseach” was modified as suggested. A sentence about the consistent advantage for ultrasound guidance without large studies reporting inferior outcomes in DIVA patients was added to the chapter and the situations with a lack of evidence (non DIVA patients and extreme emergency cases) was underlined.
Conclusion : The title was modified as suggested by the reviewer.
Reviewer 2 Report
Comments and Suggestions for Authors
Ultrasound-guided peripheral intravenous cannulation (US-PIVC) has emerged as a superior alternative to traditional “blind” techniques, which often result in high failure rates, multiple attempts, patient discomfort, and increased costs. This narrative review (2010–2025, 44 studies) highlights consistent evidence that US-PIVC improves first-attempt and overall success rates, reduces procedure time, enhances patient and staff experience, and lowers complication rates.
However, there are some revisions that would improve the manuscript:
The Results section is rich in data and references, but the text is dense. It may be useful to include summary tables of the main trials/meta-analyses (population, setting, outcomes, main findings) to make comparisons clearer.
The section on costs is interesting but focuses almost exclusively on studies from the USA and UK. It would be important to highlight the limitations in generalizing these findings to other healthcare systems (e.g., resource-limited countries, European public healthcare systems).
Author Response
We would like to thank you for your suggestions which helped us improve this manuscript.
A table summarizing the main studies cited in the manuscript was added, as requested.
Economic and Costs section. As suggested by the reviewer, we added a sentence underlying the limitations about low income countries and public continental Europe Heath systems.
Reviewer 3 Report
Comments and Suggestions for Authors
1.Title
The abstract mentions "long-term cost-effectiveness." The results section presents mixed immediate cost data. Could the abstract more clearly frame the economic argument as one of long-term system-wide savings versus short-term per-procedure cost?
2. Introduction
I do think the introduction is well written and quite good. To further strengthen it, could the authors provide a rough estimate of the "unacceptably high failure rates" for traditional methods cited in the second sentence?
3. Methods
The methods describe a systematic search strategy with independent screening, which is characteristic of a systematic review.
(a). However, the article is presented as a narrative review. Could the authors please clarify the chosen methodology? If it is a systematic review, a PRISMA flow diagram is highly recommended. If it is a narrative review, the language could be softened to reflect a more exploratory synthesis.
(b). Please include the exact search strings used for the Cochrane Library and Google Scholar, as was done for PubMed in the appendix.
4. Results
(a). In section 3.1.3, it is logically stated that improved efficiency "reduces the workload on nursing staff." Is there any data cited in the reviewed literature that quantitatively measures this reduction in nursing time or workload?
(b). The section on economic implications is very balanced. The EPIC trial noted higher immediate costs. Did any of the reviewed studies conduct a formal cost-effectiveness analysis (e.g., calculating an ICER) rather than just a cost-comparison?
(c). The technological advancements section is forward-looking. Regarding AI and robotics: did the reviewed literature identify any major barriers to the clinical implementation of these technologies (e.g., cost, regulation, need for validation studies)?
5. Discussion and Conclusion
(a). The conclusion has a strong emphasis. Could it be enhanced by adding a specific sentence addressing the main barriers to implementation (e.g., initial cost, training requirements) and suggesting strategies to overcome them? This would be highly valuable for readers.
(b). The review focuses on high-income country settings. Would the authors care to briefly speculate on the feasibility and potential challenges of implementing ultrasound-guided PIVC programs in low-resource settings?
Author Response
Thank you for your constructive suggestions.
A sentence underlining the economic impact of ultrasound-guided insertion was added to the abstract, as requested.
Introduction. As suggested by the reviewer, we added an estimate for the failure rate of traditional methods in the second sentence.
Methods. The manuscript is a narrative review, however, the reviewer 1 requested a flow diagram which was added to the manuscript. We underlined the narrative nature of the review and the absence of standardized assessment tools.
The exact search strings used for the Cochrane Library and Google Scholar, were added to the manuscript.
Results, section 3.1.3. The reduced workload is mainly measured by the time of catheterization and the reduced need for central venous catheters, as indicated in the table 1 which was added to the manuscript as requested by the reviewer 2.
Section on economic implications. Unfortunately, we did not find any study reporting a formal cost-effectiveness analysis. The reference 30 cited in the manuscript report an ICER centered on the use of a long peripheral catheter, inserted using ultrasound-guidance, but not that of ultrasound-guidance.
Regarding AI and robotics. We added a sentence underlining the fact that these robotic systems have only been used on a porcine model.
Discussion and conclusion.
A-The conclusion was modified by adding a sentence to highlight barriers to adoption of ultrasound techniques and by modifying the last sentence to propose strategies to overcome these barriers.
B-In the chapter about Economic Implications and Cost-Effectiveness, we added a sentence about the lack of studies in resource limited countries, as requested by the reviewer 2 and we believe it’s difficult to predict the challenges in these countries.
Round 2
Reviewer 1 Report
Comments and Suggestions for Authors
The authors have satisfactorily addressed all previous comments and revision requests. The revised manuscript shows clear improvements in methodological clarity, data presentation, and the integration of relevant literature in the discussion section. In its current form, the study meets the journal’s scientific standards and is suitable for publication. I recommend acceptance.
Author Response
Thank you for your help in improving the manuscript
Reviewer 2 Report
Comments and Suggestions for Authors
This narrative review examines the role of ultrasound-guided peripheral IV cannulation, analyzing 44 studies published between 2010 and 2025. The evidence consistently shows higher first-attempt success rates, shorter procedure times, improved patient and staff experience, and fewer complications compared with traditional landmark techniques, particularly in DIVA patients, pediatrics, and emergency settings. Economic data suggest potential cost savings, though findings remain limited and context-dependent. The review also discusses training requirements, guideline recommendations, and emerging technologies, while highlighting barriers to implementation and areas for future research.
Minor revisions
- The manuscript is described as a “narrative review,” yet the search and screening process (use of multiple databases, independent review by two authors) resembles a systematic approach. I recommend clarifying the methodological framework: if it is intended as a structured narrative review, this should be stated explicitly. Alternatively, adopting a simplified PRISMA flow diagram and providing more detail on inclusion/exclusion criteria would improve transparency.
- Numerical data (odds ratios, confidence intervals, p-values, percentages) are reported using different formats. For clarity and to reduce the risk of misinterpretation, I suggest standardizing the style (e.g., OR 4.25, 95% CI 2.01–8.98; p<0.001). It would also be helpful to clearly separate results related to insertion-related complications from those concerning catheter dwell time and maintenance.
- The conclusions tend to suggest that ultrasound guidance may become the standard of care for all cases of peripheral IV access. However, the strongest evidence is currently available for DIVA patients and in pediatric or emergency settings. I recommend rephrasing the conclusions more cautiously, highlighting the areas with robust evidence and those where further research is needed (e.g., non-DIVA patients, extreme emergency scenarios, and resource-limited healthcare systems).
Author Response
- The manuscript is described as a “narrative review,” yet the search and screening process (use of multiple databases, independent review by two authors) resembles a systematic approach. I recommend clarifying the methodological framework: if it is intended as a structured narrative review, this should be stated explicitly. Alternatively, adopting a simplified PRISMA flow diagram and providing more detail on inclusion/exclusion criteria would improve transparency.
- Response: We have added a clarification on the type of review in the summary and in the methodology.
- Numerical data (odds ratios, confidence intervals, p-values, percentages) are reported using different formats. For clarity and to reduce the risk of misinterpretation, I suggest standardizing the style (e.g., OR 4.25, 95% CI 2.01–8.98; p<0.001). It would also be helpful to clearly separate results related to insertion-related complications from those concerning catheter dwell time and maintenance.
- Response : Odds ratio with 95% confidence intervals and P value were reported in a standardized style, as suggested. We separate the results about the dwell time in a new paragraph.
- The conclusions tend to suggest that ultrasound guidance may become the standard of care for all cases of peripheral IV access. However, the strongest evidence is currently available for DIVA patients and in pediatric or emergency settings. I recommend rephrasing the conclusions more cautiously, highlighting the areas with robust evidence and those where further research is needed (e.g., non-DIVA patients, extreme emergency scenarios, and resource-limited healthcare systems).
- Response : the conclusion was modified as suggested (line 416) and the need for further research in non DIVA patients and extreme emergency cases has already been already added in the previous paragraph about “Identification of Areas for Future Research and Development”.